## [Decision Letter · Decision Letter 0]

8 Feb 2022

PONE-D-21-40129Provider Reported Challenges with Completing Death Certificates: A Focus Group Study Demonstrating Potential Sources of ErrorPLOS ONE

Dear Dr. Morgan,

Thank you for submitting your manuscript to PLOS ONE. After careful consideration, we feel that it has merit but does not fully meet PLOS ONE’s publication criteria as it currently stands. Therefore, we invite you to submit a revised version of the manuscript that addresses the points raised during the review process.

We look forward to receiving your revised manuscript.

Kind regards,

Benjamin P. Geisler, M.D., M.P.H., F.A.C.P., M.R.C.P. (London), F.H.M.

Academic Editor

PLOS ONE

Journal Requirements:

[This study was funded by the Northern New England Clinical Oncology Society through a Student-Led Funding grant.]

[Allie Morgan was awarded a student research grant of $2,500 from the Northern New England Clinical Oncology Society for this study. The funders had no role in study design, data collection and analysis, decision to publish, or preparation of the manuscript.]

Reviewers' comments:

Reviewer's Responses to Questions

**Comments to the Author**

1. Is the manuscript technically sound, and do the data support the conclusions?

Reviewer #1: Partly

Reviewer #2: Yes

Reviewer #3: Yes

Reviewer #4: Yes

2. Has the statistical analysis been performed appropriately and rigorously? 

Reviewer #1: Yes

Reviewer #2: N/A

Reviewer #3: Yes

Reviewer #4: N/A

3. Have the authors made all data underlying the findings in their manuscript fully available?

Reviewer #1: Yes

Reviewer #2: Yes

Reviewer #3: Yes

Reviewer #4: Yes

4. Is the manuscript presented in an intelligible fashion and written in standard English?

Reviewer #1: Yes

Reviewer #2: Yes

Reviewer #3: Yes

Reviewer #4: Yes

5. Review Comments to the Author

Reviewer #1: Dear, this is an important topic. However I have some comments:

- the background section reveals interesting papers. However the WHO has published important documents on this issue and I think it is important toe read them and see what can be worthwhile (1979 en 2020°. As I was told WHO has an e-learning on this topic. Please control it and give comments on what it can do on the education topic

- line 71: 4 groups is a low figure and 2 tot 5 participants is also a low figure. comment on that in the discussion section

- method: check for the Coreq criteria for the publication of Q research.

- line 99 it is predominantly in-patient 9 versus 5 it is almost the double. Please mention this as a weak point

- there are important papers explaining, like for cancer, that dementia is a problem: dying with or dying caused by dementia. Can you discuss that in the discussion section

- Line 141 are data available on the effectiveness of the electronic certification other than what you already mentioned?

- I was wandering that nothing came out of the results as to the problem of death certification in case of a doctor on duty, who does not know the patient and has only an (incomplete) medical file. Can you discuss this issue, because in other studies it is a problem.

- line 164-170 if doctors have different views on the importance of death certificates, it has an impact on the motivation to do some training and effort to do i properly. What is you idea about it?

- discussion section: I think a table with all the possible errors is useful

- line 273-275: is doing more autopsies a way to have better death certificates? In my clinical situation autopsies hardly are done (except in cases of criminality.

- As to spending of public funding, the epidemiology derived form death certificates, is enourmously important. It is cleat that cardiovascular and cancer deaths are far overestimated. Can you discuss the consequences of all this? What is the effect on the value of some so called evidence based figures and guidelines?

We did a research, not yet published where 92 physicians filled death certificates of 5 case vignettes: I put the results from the abstract:

91 participants filled in a death certificate for 5 cases. 90% stated the correct nature of death. The chain of death causes was correct in 30% of cases, partially correct in 45% of cases and false in 25% of cases.

Reviewer #2: This is a very sound research article. You present a concise but thorough methodology, specifically one that includes exceptional analyses of the issue at hand. Particularly your discussion of the infrastructural challenges and the issues surrounding patients with cancer was refreshing and much needed in the discipline. The Background at the beginning of the paper is somewhat brief but what’s lacking in the Background section, I think you make up for in the Discussion section. You do an excellent job at laying out the importance of your findings, strengths, weaknesses, applicable recommendations, and ideas for future exploration in the Discussion.

In lines 285 – 287 you mention “Feedback is particularly important during public 286 health crises such as the opioid epidemic and Covid-19 pandemic to ensure that deaths are 287 certified using consistent methods…” I think it might be worth adding a few sentences to this statement. No need to add in a new section of the Discussion, as we don’t want to distract from the purpose of your paper. However, as these two occurrences are highly vulnerable to the cause death determining/certification process, it may be worth adding a few sentences discussing this issue further or adding in some recent citations on the issue. Overall, this was a very solid and impactful contribution. Thank you.

Reviewer #3: Summary of the research

Cause of death certification is a timely and important topic to study. I congratulate the authors for choosing this all-important topic.

This study has been conducted to characterize the experiences of the certifiers in completing the cause of death section of death certificates. Four focus group discussions were conducted over a period of three months until the thematic saturation was reached. Three challenges leading to misclassification of causes of deaths in death certificates have been identified during the study.

The rationale for the study is clear and valid. The researchers have used a technically sound protocol and a feasible methodology to achieve the study's aims effectively.

However, I do have the following concerns about their findings and conclusions.

The international standard death certificate format that WHO recommends is also the form used in the USA. This form has two parts, namely part 1 and part 2. Part 1 has four lines for reporting conditions leading to death in a logical sequence. Part 2 is for reporting other significant conditions that may have contributed to the death. The column to the right is for reporting the approximate time interval between the onset of each of these conditions in the certificate and death.

The design of the death certificate form itself is prescriptive for the correct reporting of causes of deaths in a logical sequence. The provider’s perception of the restrictive nature of the form is due to their ignorance of the international standard death certificate format and the best practice death certification standards required to select the proper underlying cause of death.

Electronic certification can improve certification quality by avoiding common death certification errors such as illegibility, use of non-standard abbreviations, etc. However, an electronic death certificate can also be restrictive to the providers due to suggestive texts, drop-down cause lists, and other functionalities included in the format.

Therefore the solution suggested by the providers may be due to their misunderstanding of the benefits of using electronic formats. In other words, the electronic form will never reduce the misclassification of causes of deaths caused by other certification errors such as reporting ill-defined underlying causes, incorrect sequencing of causes in part 1, reporting competing causes, etc.

The appointment and providing access to death coordinators within hospitals suggested by the providers could be a meaningful solution to improve the certification practices of doctors.

Lack of training and lack of real-time feedback are important challenges for the certifying physicians, and they have been described in the published literature as well. Clinical uncertainty and certifying deaths due to cancer are already known sources of certification errors, and in this study, those issues have been reiterated from a certifier's perspective.

I also have the following concerns regarding their methodology;

1.The research question is relevant to the present context, and the rationale given in the introduction is satisfactory. However, a clear justification should be given for selecting FGD as the method of investigation.

2.Recruitment, sampling, and approach: The authors used a single case study and advocated the FGD at a single academic institution in rural New England. This selection can cause issues on the representativeness of the information obtained during FGD, therefore the generalizability of findings.

3.Strategic group bias is a recognized issue arising during organizing the FGD. The authors should explain the measures taken to minimize this bias.

4.The authors should elaborate in the methods section on the experience and the role of facilitators specifically related to the FGD.

5.The authors should explain why they use the constant comparison method as the method of thematic analysis (https://files.eric.ed.gov/fulltext/EJ1004995.pdf).

6.The authors have not discussed the limitations of the study in the discussion section.

Reviewer #4: Undoubtedly the public health significance of death reports is important and unrealized by many trainees. As such the premise for this manuscript is relevant and of significance. I liked their focus groups and interventions and breaking it down to three challenges. They focused on unwitnessed deaths and cancer patients -- it would be nice for some estimates of the proportion of this. I wish they discussed the Electronic Death Certification better, more than just real-time feedback. Is this something in pipeline for all states - what are the roadblocks?

Accuracy vs Specificity was another great question, but it didn't provide an answer which is more important or how to balance them in uncertain and cancer cases.

Finally, the autopsy requirement was an interesting area. Outside scope of paper but would love to hear reasons behind this change - cost, resources, invasiveness. Is the accuracy of death certificates worth changing this policy for?

Overall, very interesting topic with great provider identification and focus groups. I think the most useful intervention is improved education in preparing death documents with systems for real-time feedback (either on phone or electronic.) The largest interventions are a leap and would need more nuanced discussion, outside scope of this article.

6. PLOS authors have the option to publish the peer review history of their article (what does this mean?). If published, this will include your full peer review and any attached files.

Reviewer #1: No

Reviewer #2: No

Reviewer #3: **Yes: **DR. U S H Gamage

Reviewer #4: **Yes: **Adith Sekaran, MD

---

## [Author Response · Author response to Decision Letter 0]

10 Mar 2022

Reviewer #1: Dear, this is an important topic. However I have some comments:

- the background section reveals interesting papers. However the WHO has published important documents on this issue and I think it is important toe read them and see what can be worthwhile (1979 en 2020°. As I was told WHO has an e-learning on this topic. Please control it and give comments on what it can do on the education topic

We have added a statement in our discussion describing the tools available from the CDC and WHO for training providers on death certification (lines 287-89). 

- line 71: 4 groups is a low figure and 2 tot 5 participants is also a low figure. comment on that in the discussion section

Although the number of participants was relatively small, we conducted groups until saturation was reached – that is, the ideas generated during the discussions were no longer “new”, despite representation of a range of medical specialties among the participants. In the methods section we cite a paper from Guest et. al who found that 80% of themes are captured in 2-3 focus groups and 90% are captured in 3-6 focus groups in support of our decision to conduct 4 focus groups. A statement was added to the discussion to highlight that the low number of participants (2-5 per group) is a potential limitation of this study (lines 313-16). 

- method: check for the Coreq criteria for the publication of Q research.

We reviewed the Coreq criteria for qualitative research and the methods section is in line with the recommendations for the study design including a description of the theoretical framework, participant selection (purposive sampling), setting and data collection. The methods section has been revised in several places to reflect this.

- line 99 it is predominantly in-patient 9 versus 5 it is almost the double. Please mention this as a weak point

We have added this discussion to line 313: “The relatively small size of some of the focus groups and greater proportions of hospital-based participants may have limited the extent of novel ideas and quality of group discussion”. 

- there are important papers explaining, like for cancer, that dementia is a problem: dying with or dying caused by dementia. Can you discuss that in the discussion section

The particular focus of the paper was cancer (we have now revised to make this clear in line 69). The case example was cancer-related, and our grant funding was cancer-related. While we agree that dementia can also be a challenging diagnosis for providers to navigate when completing a death certificate, dementia did not come up as a theme in our focus groups and would not have been expected to given the focus of the study, and therefore falls outside of the scope of detailed discussion in the article. 

- Line 141 are data available on the effectiveness of the electronic certification other than what you already mentioned?

We discuss some additional points about electronic certification in the discussion section (beginning on line 291), but outside of the studies referenced in this section there is a paucity of data directly comparing electronic certification vs paper death certificates in the United States. 

- I was wandering that nothing came out of the results as to the problem of death certification in case of a doctor on duty, who does not know the patient and has only an (incomplete) medical file. Can you discuss this issue, because in other studies it is a problem.

This issue was brought up by participants in the second section of the results starting on line 6-162, and we have now included this in the discussion on lines 277-278. 

- line 164-170 if doctors have different views on the importance of death certificates, it has an impact on the motivation to do some training and effort to do i properly. What is you idea about it?

As a legal document that providers are required to complete, there should be an emphasis on providing all trainees either in medical school or residency with adequate training on how to accurately complete these documents. In the paragraph beginning on line 171 following table 2, there is discussion about the differing approached that providers took to resolving uncertainty in determining the cause of death. The differing approaches come from differing views on the purpose of the document rather than differing views on the importance of the document. Overall, participants found death certificates to be very important documents regardless of their approach to determining the cause of death and most supported the idea of additional training as described in the paragraph at the end of section 1: “Most targeted the lack of training or feedback, with several providers emphasizing that both training and consistent feedback are necessary for a document that is widely used for research and public health.” We have added language in the discussion on this topic on lines 225-27: “It was concerning that physicians’ views of the role and importance of death certificates to public health and to families might affect the approach they took to select a cause of death.”.

- discussion section: I think a table with all the possible errors is useful

We are not certain what the reviewer intends here but feel that the three figures provided in the results section present a reasonable summary of the study’s findings. 

- line 273-275: is doing more autopsies a way to have better death certificates? In my clinical situation autopsies hardly are done (except in cases of criminality.

We agree. In the US and elsewhere, the rate of autopsies has decreased substantially for a variety of reasons, as discussed in lines 277-81 and it seems likely that this has influenced diagnostic accuracy on the death certificate. 

- As to spending of public funding, the epidemiology derived form death certificates, is enourmously important. It is cleat that cardiovascular and cancer deaths are far overestimated. Can you discuss the consequences of all this? What is the effect on the value of some so called evidence based figures and guidelines?

We discuss the consequences of these errors in the introduction in order to describe why this study was undertaken: “In a 2010 study, deaths from cardiovascular disease in New York City were overestimated by 51% in adults ages 35-74.21 These errors have significant impacts on the public health data that underlie disease specific mortality rates, track health disparities and inform research and policy priorities. For example, in a study by Yin et al, misclassification of colon and rectal cancer deaths on the death certificate led to an inflation of the 5-year cause-specific survival rate for rectal cancer from 64.9% to 81.2%.22”

We have also included discussion on these topics on the potential impact of these errors on guidelines (lines 239-49). 

We did a research, not yet published where 92 physicians filled death certificates of 5 case vignettes: I put the results from the abstract:

91 participants filled in a death certificate for 5 cases. 90% stated the correct nature of death. The chain of death causes was correct in 30% of cases, partially correct in 45% of cases and false in 25% of cases.

Thank you for telling us about this interesting study supporting the premise that cause of death is poorly documented in death certificates. 

Reviewer #2: This is a very sound research article. You present a concise but thorough methodology, specifically one that includes exceptional analyses of the issue at hand. Particularly your discussion of the infrastructural challenges and the issues surrounding patients with cancer was refreshing and much needed in the discipline. The Background at the beginning of the paper is somewhat brief but what’s lacking in the Background section, I think you make up for in the Discussion section. You do an excellent job at laying out the importance of your findings, strengths, weaknesses, applicable recommendations, and ideas for future exploration in the Discussion.

In lines 285 – 287 you mention “Feedback is particularly important during public 286 health crises such as the opioid epidemic and Covid-19 pandemic to ensure that deaths are 287 certified using consistent methods…” I think it might be worth adding a few sentences to this statement. No need to add in a new section of the Discussion, as we don’t want to distract from the purpose of your paper. However, as these two occurrences are highly vulnerable to the cause death determining/certification process, it may be worth adding a few sentences discussing this issue further or adding in some recent citations on the issue. Overall, this was a very solid and impactful contribution. Thank you.

Thank you for this feedback. We added discussion on this beginning on line 293 to further explain the potential role that the real-time feedback of electronic medical records might have in improving accuracy without sacrificing timeliness in public health crises. 

Reviewer #3: Summary of the research

Cause of death certification is a timely and important topic to study. I congratulate the authors for choosing this all-important topic.

This study has been conducted to characterize the experiences of the certifiers in completing the cause of death section of death certificates. Four focus group discussions were conducted over a period of three months until the thematic saturation was reached. Three challenges leading to misclassification of causes of deaths in death certificates have been identified during the study.

The rationale for the study is clear and valid. The researchers have used a technically sound protocol and a feasible methodology to achieve the study's aims effectively.

However, I do have the following concerns about their findings and conclusions.

The international standard death certificate format that WHO recommends is also the form used in the USA. This form has two parts, namely part 1 and part 2. Part 1 has four lines for reporting conditions leading to death in a logical sequence. Part 2 is for reporting other significant conditions that may have contributed to the death. The column to the right is for reporting the approximate time interval between the onset of each of these conditions in the certificate and death.

The design of the death certificate form itself is prescriptive for the correct reporting of causes of deaths in a logical sequence. The provider’s perception of the restrictive nature of the form is due to their ignorance of the international standard death certificate format and the best practice death certification standards required to select the proper underlying cause of death.

Electronic certification can improve certification quality by avoiding common death certification errors such as illegibility, use of non-standard abbreviations, etc. However, an electronic death certificate can also be restrictive to the providers due to suggestive texts, drop-down cause lists, and other functionalities included in the format.

Therefore the solution suggested by the providers may be due to their misunderstanding of the benefits of using electronic formats. In other words, the electronic form will never reduce the misclassification of causes of deaths caused by other certification errors such as reporting ill-defined underlying causes, incorrect sequencing of causes in part 1, reporting competing causes, etc.

The appointment and providing access to death coordinators within hospitals suggested by the providers could be a meaningful solution to improve the certification practices of doctors.

Lack of training and lack of real-time feedback are important challenges for the certifying physicians, and they have been described in the published literature as well. Clinical uncertainty and certifying deaths due to cancer are already known sources of certification errors, and in this study, those issues have been reiterated from a certifier's perspective.

We have revised the discussion (lines 290-305) to reflect these additional perspectives. 

I also have the following concerns regarding their methodology;

1.The research question is relevant to the present context, and the rationale given in the introduction is satisfactory. However, a clear justification should be given for selecting FGD as the method of investigation.

We have added the rationale to the beginning of the methods section (lines 72-76). 

2.Recruitment, sampling, and approach: The authors used a single case study and advocated the FGD at a single academic institution in rural New England. This selection can cause issues on the representativeness of the information obtained during FGD, therefore the generalizability of findings.

We agree this is a limitation of the study and have described the difficulty with generalizability in our limitations section (lines 310-12). 

3.Strategic group bias is a recognized issue arising during organizing the FGD. The authors should explain the measures taken to minimize this bias.

We have added a sentence describing the measures taken to minimize group bias to the first paragraph of the methods section 16-(lines 87-89) and added this as a potential limitation to the discussion (lines 314-15). 

4.The authors should elaborate in the methods section on the experience and the role of facilitators specifically related to the FGD.

We have added a sentence describing the role of the facilitators to the first paragraph of the methods section (lines 80-85). 

5.The authors should explain why they use the constant comparison method as the method of thematic analysis (https://files.eric.ed.gov/fulltext/EJ1004995.pdf).

We have added a sentence explaining the selection of constant comparison analysis to the final paragraph of the methods section (lines 104-6). 

6.The authors have not discussed the limitations of the study in the discussion section.

Our discussion of limitations is now expanded to include several additional limitations including the small sample size of the study (lines 310-15). 

Reviewer #4: Undoubtedly the public health significance of death reports is important and unrealized by many trainees. As such the premise for this manuscript is relevant and of significance. I liked their focus groups and interventions and breaking it down to three challenges. They focused on unwitnessed deaths and cancer patients -- it would be nice for some estimates of the proportion of this. I wish they discussed the Electronic Death Certification better, more than just real-time feedback. Is this something in pipeline for all states - what are the roadblocks?

We have added more discussion of this in lines 290-305. One of the main roadblocks to wider implementation in the US is that each state is responsible for determining the process of death certification and therefore the US has not been able to roll out a nation-wide electronic death reporting system as many other countries have been able to. On the state level, one of the main roadblocks is funding the development and maintenance of these systems. 

Accuracy vs Specificity was another great question, but it didn't provide an answer which is more important or how to balance them in uncertain and cancer cases.

While our paper identified this as a challenge for providers, the topic and potential solutions are likely complex and we did not have space to do them justice within the paper. 

The accuracy/specificity conundrum stems from the clinician’s understanding of the role of the death certificate. Eight participants correctly view the death certificate as a legal/epidemiological tool, giving accuracy predominance. For example, the lung cancer patient who dies with one or more complications of their primary disease is properly certified as having died of complications of lung cancer, as opposed to “bronchopneumonia” or “pulmonary embolism.” Unfortunately, applying this principle too broadly leads to technically accurate, but etiologically nonsensical causes of death such as “cardiac arrest.” Six misidentify the death certificate as an element of the medical record, thus leading to the certification of sequelae of the underlying cause of death as the proximate cause. As stated above, while a legitimate question for further study, this discussion exceeds the scope of and purpose of this study beyond what is included in the discussion section.

Finally, the autopsy requirement was an interesting area. Outside scope of paper but would love to hear reasons behind this change - cost, resources, invasiveness. Is the accuracy of death certificates worth changing this policy for?

We believe that the change in Joint Commission guidelines (referenced in the paper) may underlie this change, driven primarily by cost.

Not for nothing was the term “gold standard” used in conjunction with the autopsy informing the accuracy of death certification. It is simply a myth that current diagnostics render the classic autopsy obsolete. There is an extensive literature on this subject. This flawed argument is the only medically related justification for the Joint Commission abandonment of autopsy as a quality control/quality assurance tool. The others, largely denied, are financial (hospital pathologists are loathe to spend billable time on non-reimbursed procedures that could take them hours because of their lack of autopsy experience) and hospital risk management departments inordinate fear of litigation. The latter is truly ironic in that a quality hospital autopsy prevents more litigation than it supports. The death of the hospital autopsy is largely confirmed by the fact that new hospital construction and/or renovation eliminates or excludes an autopsy suite. The expectation that the Joint Commission will reinstate a meaningful autopsy requirement is akin to expecting extruded toothpaste to be forced back into the empty tube. 

Overall, very interesting topic with great provider identification and focus groups. I think the most useful intervention is improved education in preparing death documents with systems for real-time feedback (either on phone or electronic.) The largest interventions are a leap and would need more nuanced discussion, outside scope of this article.

Thank you for this feedback.

---

## [Decision Letter · Decision Letter 1]

3 May 2022

Provider Reported Challenges with Completing Death Certificates: A Focus Group Study Demonstrating Potential Sources of Error

PONE-D-21-40129R1

Dear Dr. Morgan,

We’re pleased to inform you that your manuscript has been judged scientifically suitable for publication and will be formally accepted for publication once it meets all outstanding technical requirements.

Kind regards,

Benjamin P. Geisler, M.D., M.P.H., F.A.C.P., M.R.C.P. (London), F.H.M.

Academic Editor

PLOS ONE

Additional Editor Comments (optional):

Reviewers' comments:

Reviewer's Responses to Questions

**Comments to the Author**

1. If the authors have adequately addressed your comments raised in a previous round of review and you feel that this manuscript is now acceptable for publication, you may indicate that here to bypass the “Comments to the Author” section, enter your conflict of interest statement in the “Confidential to Editor” section, and submit your "Accept" recommendation.

Reviewer #1: All comments have been addressed

Reviewer #2: All comments have been addressed

Reviewer #3: All comments have been addressed

2. Is the manuscript technically sound, and do the data support the conclusions?

Reviewer #1: Yes

Reviewer #2: Yes

Reviewer #3: Yes

3. Has the statistical analysis been performed appropriately and rigorously? 

Reviewer #1: Yes

Reviewer #2: N/A

Reviewer #3: Yes

4. Have the authors made all data underlying the findings in their manuscript fully available?

Reviewer #1: Yes

Reviewer #2: Yes

Reviewer #3: Yes

5. Is the manuscript presented in an intelligible fashion and written in standard English?

Reviewer #1: Yes

Reviewer #2: Yes

Reviewer #3: Yes

6. Review Comments to the Author

Reviewer #1: You answered succesfully to my remarks and I hope that this paper will have an interesting impact. About line 69 dying with or dying caused by it was not my goal to comment on dementia (it was just an example of my own research). It is important to make the difference between the two concepts.

Reviewer #2: Thank you for your revisions. Your additions add nicely to the quality of your study and help to position your findings in contemporary contexts.

Reviewer #3: You have addressed all my previous comments adequately. Thanks for choosing such an interesting topic for your study.

7. PLOS authors have the option to publish the peer review history of their article (what does this mean?). If published, this will include your full peer review and any attached files.

Reviewer #1: **Yes: **De Lepeleire Jan

Reviewer #2: No

Reviewer #3: No

---

## [Editor Report · Acceptance letter]

11 May 2022

PONE-D-21-40129R1 

Provider Reported Challenges with Completing Death Certificates: A Focus Group Study Demonstrating Potential Sources of Error 

Dear Dr. Morgan:

I'm pleased to inform you that your manuscript has been deemed suitable for publication in PLOS ONE. Congratulations! Your manuscript is now with our production department. 

Kind regards, 

on behalf of

Dr. Benjamin P. Geisler 

Academic Editor

PLOS ONE